# A Negative Body Image among Adolescent and Young Adult (AYA) Cancer Survivors: Results from the Population-Based SURVAYA Study

**DOI:** 10.3390/cancers14215243

**Published:** 2022-10-26

**Authors:** Laura M. H. Saris, Carla Vlooswijk, Suzanne E. J. Kaal, Janine Nuver, Rhodé M. Bijlsma, Tom van der Hulle, Mathilde C. M. Kouwenhoven, Jacqueline M. Tromp, Roy I. Lalisang, Monique E. M. M. Bos, Winette T. A. van der Graaf, Olga Husson

**Affiliations:** 1Department of Psychosocial Research and Epidemiology, Netherlands Cancer Institute, 1066 CX Amsterdam, The Netherlands; 2Clinical Sciences for Health Professionals, Program in Clinical Health Sciences, University Medical Center Utrecht, Utrecht University, 3584 CS Utrecht, The Netherlands; 3Research and Development, Netherlands Comprehensive Cancer Organisation, 3511 DT Utrecht, The Netherlands; 4Department of Medical Oncology, Radboud University Medical Center, 6525 GA Nijmegen, The Netherlands; 5Dutch AYA ‘Young and Cancer’ Care Network, Netherlands Comprehensive Cancer Organisation, 3511 DT Utrecht, The Netherlands; 6Department of Medical Oncology, University Medical Center Groningen, 9713 GZ Groningen, The Netherlands; 7Department of Medical Oncology, University Medical Center, 3584 CX Utrecht, The Netherlands; 8Department of Medical Oncology, Leiden University Medical Center, 2333 ZA Leiden, The Netherlands; 9Department of Neurology, Amsterdam UMC, Amsterdam University Medical Centers, Location VUmc, 1081 HV Amsterdam, The Netherlands; 10Department of Medical Oncology, Amsterdam University Medical Centers, 1105 AZ Amsterdam, The Netherlands; 11Division Medical Oncology, Department of Internal Medicine, GROW-School of Oncology and Reproduction, Maastricht UMC+ Comprehensive Cancer Center, Maastricht University Medical Center+, 6229 HX Maastricht, The Netherlands; 12Department of Medical Oncology, Erasmus MC Cancer Institute, Erasmus University Medical Center, 3015 GD Rotterdam, The Netherlands; 13Department of Medical Oncology, Netherlands Cancer Institute-Antoni van Leeuwenhoek, 1066 CX Amsterdam, The Netherlands; 14Division of Clinical Studies, Institute of Cancer Research, Royal Marsden NHS Foundation Trust, London SM2 5NG, UK; 15Department of Surgical Oncology, Erasmus MC Cancer Institute, Erasmus University Medical Center, 3015 GD Rotterdam, The Netherlands

**Keywords:** body image, adolescents and young adults, cancer survivorship, population-based research

## Abstract

**Simple Summary:**

Adolescent and young adult (AYA) cancer survivors diagnosed with cancer between ages 18–39 years often experience negative body changes, such as scars, amputation, hair loss, disfigurement, body weight changes, skin buns, and physical movement limitations. A negative body image could have negative implications for the self-esteem, self-identity, and social relationships of AYAs. Despite the possible long-term effects of cancer on body image, within the AYA literature, limited studies focus on AYA cancer survivors in a quantitative way. Therefore, the aim of our population-based cross-sectional study was to examine the prevalence, and association of a negative body image with sociodemographic, clinical, and psychosocial factors, among AYA survivors 5–20 years after diagnosis. Raising awareness and integrating supportive care for those who experience a negative body image into standard AYA survivorship care is warranted. Future longitudinal research could help to identify when and how this support for AYA survivors can be best utilized.

**Abstract:**

Adolescent and young adult (AYA) cancer survivors (18–39 years at diagnosis) often experience negative body changes such as scars, amputation, and disfigurement. Understanding which factors influence body image among AYA survivors can improve age-specific care in the future. Therefore, we aim to examine the prevalence, and association of a negative body image with sociodemographic, clinical, and psychosocial factors, among AYA cancer survivors (5–20 years after diagnosis). A population-based cross-sectional cohort study was conducted among AYA survivors (5–20 years after diagnosis) registered within the Netherlands Cancer Registry (NCR) (SURVAYA-study). Body image was examined via the EORTC QLQ-C30 and QLQ-SURV100. Multivariable logistic regression models were used. Among 3735 AYA survivors who responded, 14.5% (range: 2.6–44.2%), experienced a negative body image. Specifically, AYAs who are female, have a higher Body Mass Index (BMI) or tumor stage, diagnosed with breast cancer, cancer of the female genitalia, or germ cell tumors, treated with chemotherapy, using more maladaptive coping strategies, feeling sexually unattractive, and having lower scores of health-related Quality of Life (HRQoL), were more likely to experience a negative body image. Raising awareness and integrating supportive care for those who experience a negative body image into standard AYA survivorship care is warranted. Future research could help to identify when and how this support for AYA survivors can be best utilized.

## 1. Introduction

Although cancer is a disease primarily affecting older adults, each year around 3900 adolescents and young adults (AYAs) aged 18–39 years in the Netherlands are diagnosed with cancer for the first time [1]. AYAs are recognized as a distinct population within oncology, since they face unique challenges given the complex phase of life, including many physical, emotional, and social transitions [2,3,4,5,6,7,8]. Important and complex age-related developmental milestones need to be achieved, including forming their own identity and a positive body image; establishing autonomy, responsibility, and independence; finishing education and starting a career; beginning a relationship and having children [9,10]. The cancer diagnosis and treatment(s) can disrupt these physical, cognitive, and psychosocial developmental milestones for AYAs, which can lead to a reduced health-related quality of life (HRQoL) [4,6,10,11]. Since the overall five-year survival rate for AYAs has improved to 80%, many AYAs have a long life ahead after their cancer diagnosis [12]. Therefore, achieving and maintaining optimal HRQoL is an important aspect of AYAs [4,13,14].

A major concern for AYAs potentially affecting their HRQoL, are problems with body image [3,4,7,10,13,15]. For cancer patients, body image is an important multidimensional and complex concept connected to multiple aspects of cancer and its treatments [13,16,17]. Body image involves positive and negative perceptions, thoughts, feelings, and behaviors about the entire body and its functioning [13,16,17]. Cancer and related treatments can cause temporary or permanent body changes, which may negatively affect body image [13,16,18,19]. AYAs describe that their body is greatly affected by cancer and the treatment(s), with negative changes in body image such as scars, amputation, hair loss, disfigurement, changes in body weight, skin burns, and limitations in physical movements [7,18,19]. Some AYAs describe that they no longer feel in control of their body after cancer, and see their body as a threat to their health and functioning, or as a source of discomfort [18]. Adding to this, a negative body image could have negative implications for the self-esteem, self-identity, and social relationships of AYAs [7,11,19,20].

Different studies have supported the need of understanding the complexity of body image within the AYA population [4,11,13,21]. A recent scoping review by Vani et al. [13] showed that existing quantitative studies are mainly addressing body image from the perspective of tumor histology, focusing on specific cancer groups, rather than from an AYA-specific perspective [13,14]. The available studies with an age-specific focus have a qualitative design, indicating that currently little is known about the prevalence and what factors are associated with a negative body image [21,22,23,24]. Moreover, despite the possible long-term effects of cancer on body image, within the AYA literature limited attention has been given to long-term AYA cancer survivors (>five years after diagnosis) [4,13].

To raise understanding, the aim of this population-based study is to assess the prevalence of a negative body image among AYA survivors 5–20 years after diagnosis. Furthermore, we aim to examine the association between sociodemographic, clinical, and psychosocial factors and a negative body image among AYA survivors 5–20 years after diagnosis. Understanding who is at risk for a negative body image and why will help to improve age-specific care regarding body image issues for AYA survivors in the future.

## 2. Materials and Methods

### 2.1. Setting and Population

This population-based cross-sectional cohort study was performed among AYA cancer survivors (18–39 years old at the time of diagnosis) registered within the Netherlands Cancer Registry (NCR): the SURVAYA study. The SURVAYA study (Health-related quality of life and late effects among SURVivors of cancer in Adolescence and Young Adulthood) was conducted in the Netherlands Cancer Institute and all University Medical Centers in the Netherlands. The SURVAYA study was approved by the Netherlands Cancer Institute Institutional Review Board (IRB-IRBd18122) and was registered within clinical trial registration (NCT05379387). The Netherlands Cancer Registry (NCR) was used to select the AYA survivors for the SURVAYA study and the AYA survivors from the SURVAYA study who at least answered half of the ten items on body image were included in this secondary analysis on body image.

### 2.2. Data Collection

Data for the SURVAYA study were collected between May 2019 and June 2021 within PROFILES (Patient Reported Outcomes Following Initial treatment and Long term Evaluation of Survivorship) [14], which is a registry for the study of the physical and psychosocial impact of cancer and its treatments and is directly linked to clinical data from the NCR [14]. The NCR routinely collects data on tumor characteristics and patients’ background characteristics at the primary diagnosis. Details on data collection for the SURVAYA study have previously been described [25]. 

### 2.3. Study Measures

Factors potentially associated with a negative body image, based on the literature [13,26,27,28] and used in this study, were: age (at questionnaire), gender, relationship status, level of education, tumor type, tumor stage, treatment, time since diagnosis, physical activity level, Body Mass Index (BMI), Coping style, HRQoL, and sexual attractiveness.

Sociodemographic data, like age at questionnaire, gender (male/female), relationship status (partner yes/no), and level of education (no or primary school (low), secondary school (intermediate), college or university (high)) were collected by self-report in the questionnaire. Clinical data including tumor type, tumor stage (I–IV), treatment (surgery (local or organ), chemotherapy, radiotherapy, hormone therapy, targeted therapy, or stem cell transplantation), and date of diagnosis were available from the NCR. 

The level of physical activity was assessed with items derived from the validated European Prospective Investigation into Cancer (EPIC) Physical Activity Questionnaire [29]. A total level of physical activity was calculated by summing up all hours/week of all activities (walking, bicycling, gardening, housekeeping, and sports) [29]. To include an estimate of intensity, metabolic equivalent intensity values (MET) were assigned to each activity according to the compendium of physical activity, and a total physical activity in MET hours/week was calculated [30]. BMI was calculated with the self-reported body height and weight and categorized into underweight (BMI < 18.5), normal weight (18.5 < BMI < 25), overweight (25 < BMI < 30), and obesity (BMI > 30) [31]. 

The Cognitive Emotion Regulation Questionnaire (CERQ) was part of the questionnaire and used to identify which cognitive coping strategies AYA survivors use when experiencing negative events or situations [32,33,34]. The identified coping styles were dichotomized into two general categories: adaptive (acceptance, positive refocusing, refocus on planning, positive reappraisal, putting into perspective) and maladaptive (self-blame, rumination, catastrophizing, and blaming others) [32,34]. A higher score (0–50 for adaptive and 0–40 for maladaptive) indicates an AYA survivor is using the coping style more frequently in response to a negative event [32]. 

HRQoL was measured by the functional scales and global health status from the European Organization for Research and Treatment of Cancer Quality of Life Questionnaire (EORTC QLQ-C30), in which higher scores represent a higher level of functioning compared to a lower score [35]. Furthermore, a one-scale item “Have you been feeling less sexually attractive as a result of your disease or treatment?” was used to describe sexual attractiveness.

Body image was assessed with ten items from the EORTC QLQ-C30 [35] and an extended version of the EORTC HRQoL cancer survivorship core questionnaire (QLQ-SURV100) [36]. All ten items were scored on a four-point Likert scale. To describe each item separately, a score of three points or higher was considered and dichotomized as having a negative body image. Furthermore, to assess the prevalence of a negative body image, a total score of body image was calculated according to the EORTC scoring guidelines for each AYA survivor, scoring from 0–100 [37]. Higher scores indicate more symptoms of a negative body image. A total body image score of at least one standard deviation above the mean of all body image scores (cut-off: ≥38.15) was considered and dichotomized as having a negative body image [37]. 

### 2.4. Data Analysis

The study population was described with means (including standard deviations) and frequencies (with percentages), stratified by body image (“negative body image” vs. “not a negative body image”). Chi-square and independent *t*-tests were used to assess differences between the group who answered at least half of the items on body image and those who did not. 

To examine the prevalence, a proportion of how many AYA survivors experienced a negative body image was calculated, for a total of 10 items and for each item separately. To assess the association between potential factors and a negative body image, univariable logistic regression models were fitted. Multivariable logistic regression analyses were conducted by including all independent significant associations from the univariable model (*p*-value < 0.1). Multicollinearity in the multivariable model was explored by using the tolerance statistic (*p*-value < 0.1 indicate multicollinearity), variance inflation factor (VIF > 10 indicates multicollinearity), and variance proportions (proportions on the same eigenvalue ≥ 0.7 indicate multicollinearity). A sensitivity analysis was conducted to explore the effect of the item on scars (this item was not applicable to everyone) on the total prevalence of a negative body image and the logistic regression model.

If items on the EORTC-scales for body image and HRQoL were missing, and at least half of the items from that scale were answered, the scale scores were calculated by mean case analysis [37]. If less than half of the items from the scales were answered, the scale score was set to missing [37]. For all other variables, missing items were set to missing. For categorical variables with more than 30 missing values, an extra category was created with the label missing values. For continuous variables, missing data were handled by pairwise deletion.

All analyses were conducted using IBM SPSS Statistics for Windows, version 28 (IBM Corp., Armonk, NY, USA) and all tests were two-sided with a *p*-value < 0.05 for statistical difference. 

## 3. Results

### 3.1. Patient and Tumor Characteristics

In total, in the SURVAYA study *n* = 4010 AYA survivors completed the questionnaire, resulting in an overall response rate of 36%. In this secondary analysis on body image *n* = 3735 eligible, AYA survivors were included. The differences in population characteristics of AYA survivors who answered at least half of the ten items on body image (eligible) and who did not (not eligible), are displayed in Table 1. Differences between these groups were seen in tumor type and chemotherapy received (yes/no).

Table 2 describes the characteristics of the included AYA survivors. The mean age of these AYA survivors at the time of the questionnaire was 44.5 (SD 7.5) years with a mean time since diagnosis of 12.4 (SD 4.5) years. The most common diagnoses were breast cancer (23.7%), germ-cell tumors (17.5%), cancer of the female genitalia (10.9%), melanomas (6.9%), and thyroid cancer (6.1%). Most AYA survivors were female (61.0%), had a partner when completing the questionnaire (83.1%), and were diagnosed at stage I (42.7%).

### 3.2. Body Image

Overall, 541 (14.5%) AYA survivors experienced a negative body image (Table 3 and Figure 1). Almost half (44.2%) reported that their body did not feel complete. One out of six AYA survivors reported that they felt older than their age (16.8%), were dissatisfied with their physical appearance (16.2%), and felt less masculine/feminine due to cancer or its treatment (15.1%). A small percentage of AYA survivors avoided people because of how they felt about their physical appearance (2.6%) or felt embarrassed about their physical appearance (8.0%).

### 3.3. Logistic Regression

Univariable logistic regression models showed that females, AYA survivors without a partner, those with a lower level of education, and higher tumor stage were significantly more likely to experience a negative body image (Table 4). Furthermore, tumor type, chemotherapy, radiotherapy, hormone therapy, targeted therapy, BMI, maladaptive coping style, sexual attractiveness, the functional scales of HRQoL, and global health status were all independently associated with a negative body image.

The multivariable analysis showed that gender, tumor type, tumor stage, chemotherapy, BMI, maladaptive coping style, role functioning, emotional functioning, cognitive functioning, social functioning, global health status (HRQoL), and sexual attractiveness were associated with a negative body image (Table 4). Females were more likely to experience a negative body image compared to men (Odds ratio (OR) = 3.79 [95%Confidence interval (CI): 2.49–5.77]). AYA survivors with breast cancer (OR = 2.51 [95%CI: 1.19–5.28]), cancer of the female genitalia (OR = 3.15 [95%CI: 1.59–6.24]), and germ cell tumors (OR = 2.50 [95%CI: 1.11–5.65]) experienced a negative body image more often compared to AYA survivors with a melanoma. Stage II (OR = 1.49 [95%CI: 1.07–2.10]) and III (OR = 1.72 [95%CI: 1.13–2.61]) disease were associated with greater odds of having a negative body image compared to stage I. AYA survivors who received chemotherapy had lower odds (OR = 0.68 [95%CI: 0.47–0.99]) for having a negative body image compared to those who received no chemotherapy. Furthermore, AYA survivors with overweight (OR = 1.70 [95%CI: 1.29–2.24]) or obesity (OR = 3.69 [95%CI: 2.66–5.13]) more likely experienced a negative body image compared to those with a normal weight. AYA survivors who more often used a maladaptive coping style (higher scores) were more likely to experience a negative body image (OR = 1.10 [95%CI: 1.06–1.13]) compared to those who used less maladaptive coping styles. AYA survivors with higher levels of role functioning (OR = 0.99 [95%CI: 0.99–1.00]), emotional functioning (OR = 0.97 [95%CI: 0.97–0.98]), cognitive functioning (OR = 0.99 [95%CI: 0.99–1.00]), social functioning (OR = 0.99 [95%CI: 0.98–1.00]), and global health status (OR = 0.98 [95%CI: 0.97–0.99]) were less likely to experience a negative body image. AYA survivors who felt sexually unattractive, had greater odds (OR = 3.73 [95%CI: 2.50–5.57]) for a negative body image.

When leaving out the item on scars, the multivariable model showed no differences in associated factors with a negative body image (Appendix A).

## 4. Discussion

This large population-based cross-sectional cohort study showed that a negative body image remains a problem up to 20 years after cancer diagnosis for almost 15% (range: 2.6–44.2%) of the AYA survivors. Multiple factors were associated with a long-term negative body image, including sociodemographic (gender), clinical (tumor type, chemotherapy, stage, and BMI), and psychosocial factors (maladaptive coping style, sexual attractiveness, role functioning, emotional functioning, cognitive functioning, social functioning, and global health status).

Among (long-term) breast cancer survivors (not AYA specific) the prevalence of a negative body image is reported between 15–33% [38,39], which is in line with our results. When comparing our results to AYA-specific literature, Vani et al. [13] reported a prevalence of 17–63% among AYA patients. However, the patients participating in the study of Vani et al. [13] were diagnosed more recently and literature shows that AYAs experience more body image concerns when they were on treatment compared to the (first) years after treatment [18,19,40,41]. Furthermore, most improvements in HRQoL and body image occur in the first two years after diagnosis and remain relatively stable thereafter [42,43]. This might explain the lack of an association between time since diagnosis and body image in this study, as we only focused on long-term survivors (5–20 years later). 

Most literature shows that female (AYA) patients and survivors are more likely to have a negative body image compared to males [13,17,19,44]. This is in line with our results, which show that females are more likely to experience a negative body image. According to Zucchetti et al. [19], female survivors report more fears of gaining weight and worries related to their physical appearance than males. Also, according to DeFrank et al. [43] female patients place more emphasis on appearance and sexual-related side effects than male patients.

In addition, in line with Vani et al. [13] who reported body changes reduced AYAs sexual attractiveness, our study shows a negative body image was associated with feeling sexually unattractive. This is supported by the study of Graugaard et al. [44], which showed that the risk of body image problems and attractiveness issues increased when sexual problems were present.

Similar to other studies [15,43,44], our findings show that a negative body image is associated with tumor type. This association might be explained by the unique treatments and subsequent side effects that come with various types of cancer [43,44]. For some cancers, like breast cancer and melanomas, physical alterations may be more visible and thus more disturbing [43,44]. Our study shows that breast, female genitalia, and germ cell tumor survivors were more at risk for a negative body image than melanoma survivors. When cancer affected the breast or reproductive organs, the AYA survivors might feel less attractive or less feminine/masculine, negatively affecting their body image [45,46]. Furthermore, it could be hypothesized that body image (and sexual attractiveness) are affected by the changed hormone (estrogen or testosterone) levels due to cancer treatment, especially in cancer of the reproductive organs [46,47,48]. Although in line with most studies, our results are in incongruence with the findings of Graugaard et al. [44], showing genital cancer patients were significantly less at risk for a negative body image than melanoma patients. A possible explanation for this difference is that Graugaard et al. [44] pooled gender together for cancer of the genitalia and we included a wider range of cancer types.

The association between higher tumor stages and a negative body image was also found in other studies and might relate to patients with advanced tumors (larger size or higher stage) often undergoing more invasive or multimodal treatments and being more predisposed to visible scarring or disfigurement [43,49,50]. Furthermore, it is not entirely surprising to find a negative body image associated with BMI and maladaptive coping styles, since these associations were also found in the general population [50,51,52,53,54]. Overweight and obese individuals often report a negative body image and have more weight and shape concerns than individuals with a normal weight [55]. Individuals who are using more maladaptive coping styles, such as avoidant coping, are more likely to have a negative body image and believe their personal worth is influenced by their physical appearance [51]. Although these associations are most likely to be not specific to the cancer population, this does suggest that AYA survivors with a higher BMI or maladaptive coping styles are more at risk for having a negative body image. Consistent with previous findings relationship status, level of education, age at questionnaire, and time since diagnosis were not associated with a negative body image [17,26,40,43,44,56]. Although having a partner is not associated with body image, according to Kowalczyk et al. [47] the relationship quality and (partner) support level might be associated with a better body image. Furthermore, in contrast to previous studies, our study showed no association between the level of physical activity and body image [13,26]. This difference may have been caused by the high reported levels of physical activity in our study which are potentially caused by an overestimation of reality [57]. Although the level of physical activity might be overestimated due to a measurement error, it did give the opportunity to rank AYA survivors according to physical activity level [57,58].

In contrast to others reporting no association between treatments and body image [43,56,59], we found an association between a negative body image and chemotherapy. Although chemotherapy is known to cause side effects such as hair loss, weight gain, and sexual dysfunction, our study showed that AYA survivors who received chemotherapy were less likely to experience a negative body image a long time after diagnosis [43]. Chemotherapy side effects are mostly temporary and it could be hypothesized that AYA survivors who received chemotherapy might have received less invasive surgeries [43]. 

Considering the limited amount of (quantitative) data available on body image among AYA survivors, this study provides an important contribution and insight into the long-term impact of cancer and several related factors with a negative body image. Another strength of this study is the population-based design including a large number of patients. Furthermore, participants could complete the questionnaires when, where, and how (online or on paper) they wanted. Although the item on dissatisfaction with scars was not applicable for some AYA survivors, the sensitivity analysis showed that this item did not change the measured concept of body image, since the same associated factors were identified through both models.

The present study has also some limitations that should be mentioned. Since the clinical data were collected through the NCR, we only have information on the primary diagnosis and treatment up to one year after diagnosis. Therefore, it is unknown whether survivors received other treatments in the 5–20 years after diagnosis which could affect body image outcomes. Over time treatments and surgical techniques have been improved and some became less invasive. However, in this study, we were not able to look at these differences over time since the data of the NCR are not detailed enough. Furthermore, we were only able to provide general information on treatments, instead of looking at details (such as the type of stem cell transplantation or included reconstructive surgeries). The effect of immunotherapy on body image among AYA survivors also remains unknown because we were not able to include this relatively new form of therapy in our analysis. Since immunotherapy is often used in current cancer treatments of AYAs, it is important to include this in future research. In addition, the cross-sectional design limits the determination of causality. Since we used secondary data from the SURVAYA study, body image was measured with items from the EORTC QLQ-C30 and QLQ-SURV100 which are cancer-generic instruments. There are several measurement tools, such as the body image scale (BIS), developed and validated more specifically for measuring the concept of body image among cancer patients [18,60,61]. However, the content of the items of the QLQ-SURV100 was comparable to the BIS items. The cancer-generic tools made it impossible to compare our data on the prevalence of body image problems to the general population, leaving it unknown whether part of the body image problems might be caused by the specific life stage of AYAs. More general limitations of the SURVAYA study have previously been described [25].

## 5. Implications for Clinical Practice and Future Research

Overall, this study highlights the need for healthcare professionals to open the dialogue on body image and address body image as a standard topic in AYA survivorship care [13]. Since many long-term effects can be mitigated through targeted surveillance and early intervention, healthcare professionals who provide (multidisciplinary) age-specific AYA care play a key role in the early identification and intervention of body image problems [13,19,43,62]. The results of this study can help healthcare professionals identify which AYA survivors are more at risk for having a negative body image and can help to start discussing body image with AYA survivors. Furthermore, the AYA Healthcare Network has developed several tools contributing to the provision of age-specific care, including an anamnesis tool based on the self-identified needs to facilitate conversations between healthcare professionals and AYAs [63]. One of the themes in this anamnesis is body image [63]. The AYA anamnesis is currently implemented in all centers providing age-specific care in the Netherlands [63]. When body image problems are identified, healthcare professionals could provide tips and resources for managing body changes, and can help AYA survivors by strengthening their self-image and recommending appropriate interventions to improve their body image [13,19,40,43]. 

Given the complexity of body image, it is important that those interventions meet the AYA survivors’ individual needs and (age-specific) preferences [13,64]. AYAs should be encouraged to engage with supportive others (such as family, partner, or other cancer survivors) about their body experiences [40,65]. Furthermore, in the literature, different types of pre-post intervention/programs are described that have positive outcomes on the body image of AYA patients: individual/group interventions, in-person/online, and with single/multiple sessions, focusing on body image alone or as part of an intervention for different psychosocial issues (e.g., cognitive behavioral therapy or psycho-education) [13,38,64,65]. 

Future research with longitudinal designs could allow us to assess experienced body image over time to determine the best time to intervene for those AYA survivors with a negative body image. Furthermore, available interventions for AYA patients should be validated among AYA survivors or adapted when needed [11,13].

## 6. Conclusions

A negative body image can be a long-term issue for AYA cancer survivors, specifically for females, those with a higher BMI or tumor stage, and those diagnosed with breast cancer, cancer of the female genitalia, or germ cell tumors. Also, AYA survivors who did not undergo chemotherapy, who use more maladaptive coping strategies, who feel sexually unattractive, or who have lower HRQoL are more likely to experience long-term body image problems. Healthcare professionals who provide (multidisciplinary) age-specific AYA care play a key role in the early identification and intervention of body image problems. Raising awareness and integrating supportive care for those who experience a negative body image into standard AYA survivorship care is warranted. Additionally, further (longitudinal) research could help to improve age-specific care for AYA survivors focusing on body image.

## Figures and Tables

**Figure 1 cancers-14-05243-f001:**
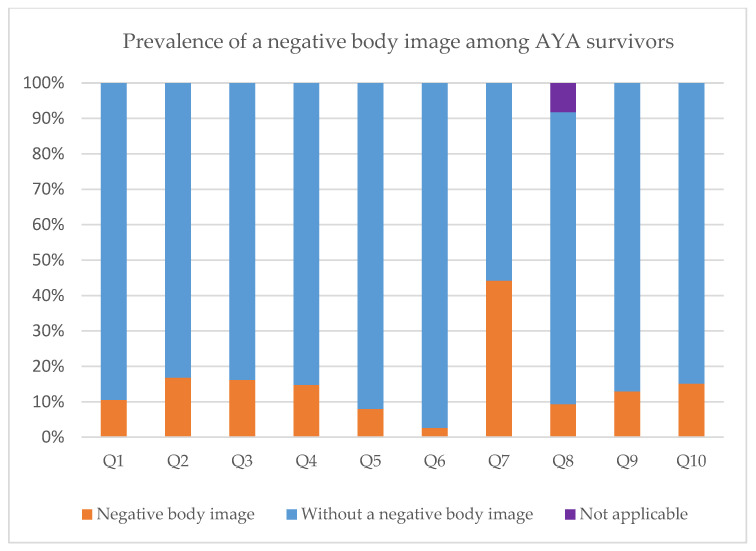
Prevalence of a negative body image among AYA survivors: Q1. Have you felt unattractive? Q2. Have you felt older than your age? Q3. Have you been dissatisfied with your physical appearance? Q4. Have you felt that you could not trust your body? Q5. Have you felt embarrassed about your body? Q6. Did you avoid people because of the way you felt about your appearance? Q7. Did your body feel complete? Q8. Have you been dissatisfied with the appearance of the scars? (Participants could also answer ‘not applicable’) Q9. Did you judge your physical appearance more negatively since the diagnosis and treatment of cancer? Q10. Have you felt less masculine/feminine as a result of your illness or treatment since the diagnosis and treatment of cancer?

**Table 1 cancers-14-05243-t001:** Differences in characteristics of the included and excluded AYA survivors.

Population Characteristics		Included AYA Survivors ^1^		Excluded AYA Survivors ^2^		*p*-Value ^3^
		n = 3735		n = 275		
		n	%	n	%	
Age at questionnaire—Mean (SD)		44.5 (7.5)		44.0 (7.2)		0.242
Gender	Male	1456	39.0	93	33.8	0.090
	Female	2279	61.0	182	66.2	
Type of Cancer	Melanoma	258	6.9	32	11.6	**0.034**
	Head and neck	115	3.1	9	3.3	
	Colon and rectal	76	2.0	6	2.2	
	Digestive tract other ^4^	30	0.8	1	0.4	
	Breast	885	23.7	59	21.5	
	Female genitalia	407	10.9	38	13.8	
	Thyroid gland	228	6.1	20	7.3	
	Central nervous system	142	3.8	8	2.9	
	Bone and soft tissue sarcoma	165	4.4	7	2.5	
	Germ cell tumor	653	17.5	39	14.2	
	Lymphoid hematological malignancies	555	14.9	36	13.1	
	Myeloid hematological malignancies	140	3.7	8	2.9	
	Other ^5^	81	2.2	12	4.4	
Treatments ^6^	Surgery organ (yes) ^7^	2483	66.6	172	62.5	0.175
	Surgery local (yes) ^8^	537	14.4	49	17.8	0.121
	Chemotherapy (yes)	2104	56.4	135	49.1	**0.019**
	Radiotherapy (yes) ^9^	1779	47.7	128	46.5	0.716
	Hormone therapy (yes)	455	12.2	29	10.5	0.418
	Targeted therapy (yes)	289	7.7	18	6.5	0.470
	Stem cell transplantation (yes)	135	3.6	7	2.5	0.353
Tumor stage	I	1595	42.7	131	47.6	0.358
	II	991	26.5	72	26.2	
	III	535	14.3	38	13.8	
	IV	171	4.6	8	2.9	
	Unknown	443	11.9	26	9.5	
Age at diagnosis—Mean (SD)		31.6 (5.9)		31.6 (5.5)		0.993
Time since diagnosis—Mean (SD)		12.4 (4.5)		11.9 (4.6)		0.057
	<10 years	1274	34.1	111	40.4	0.109
	10–15 years	1307	35.0	87	31.6	
	>15 years	1154	30.9	77	28.0	

^1^ AYA survivors who completed at least half of the items on body image and were included in the statistical analysis. ^2^ AYA survivors who completed less than half of the items on body image were excluded from the statistical analysis. ^3^ The bold p-values show a statistically significant difference (*p*-value < 0.05) between the two groups. ^4^ Digestive tract and other includes the esophagus, stomach, and small intestine. ^5^ Other includes respiratory, male genitalia (penis, prostate), urinary tract, tumor with other localizations, endocrine glands, eye, neuroblastoma, paraganglioma. ^6^ The treatments were received at primary diagnosis (missing *n* = 4 for included AYA survivors because the NCR did not provide therapy registration for them). ^7^ Organ surgery is defined as the complete resection of the affected organ. ^8^ Local surgery is defined as resection of the tumor/metastasis only. ^9^ Radiotherapy includes radiotherapy for primary tumors and metastases at primary diagnosis. *n* = number of AYA survivors, Mean (SD) = mean and standard deviation.

**Table 2 cancers-14-05243-t002:** Sociodemographic, clinical, and psychosocial characteristics of the included AYA survivors.

Population Characteristics		Total Included AYA Survivors ^1^		AYA Survivors with a Negative Body Image		AYA Survivors without a Negative Body Image		Missing
		*n* = 3735		*n* = 541		*n* = 3194		
		*n*	%	*n*	%	*n*	%	*n*
Sociodemographic characteristics								
Age at questionnaire—Mean (SD)		44.5 (7.5)		44.2 (7.3)		44.6 (7.5)		
Gender	Male	1456	39.0	94	6.5	1362	93.5	
	Female	2279	61.0	447	19.6	1832	80.4	
Partner (yes)		3107	83.4	395	12.7	2710	87.3	13
Level of education	Low	24	0.6	6	25.0	18	75.0	8
	Intermediate	1601	42.9	277	17.3	1322	82.7	
	High	2106	56.4	257	12.2	1847	87.8	
Clinical characteristics								
Type of Cancer	Melanoma	258	6.9	18	7.0	240	93.0	
	Head and neck	115	3.1	12	10.4	103	89.6	
	Colon and rectal	76	2.0	8	10.5	68	89.5	
	Digestive tract other ^2^	30	0.8	6	20.0	24	80.0	
	Breast	885	23.7	191	21.6	694	78.4	
	Female genitalia	407	10.9	92	22.6	315	77.4	
	Thyroid gland	228	6.1	26	11.4	202	88.6	
	Central nervous system	142	3.8	22	15.5	120	84.5	
	Bone and soft tissue sarcoma	165	4.4	20	12.1	145	87.9	
	Germ cell tumor	653	17.5	38	5.8	615	94.2	
	Lymphoid hematological malignancies	555	14.9	76	13.7	479	86.3	
	Myeloid hematological malignancies	140	3.7	21	15.0	119	85.0	
	Other ^3^	81	2.2	11	13.6	70	86.4	
Treatments ^4^	Surgery organ (yes)	2483	66.6	368	14.8	2115	85.2	4
	Surgery local (yes)	537	14.4	69	12.8	468	87.2	4
	Chemotherapy (yes)	2104	56.4	342	16.3	1762	83.7	4
	Radiotherapy (yes) ^5^	1779	47.7	299	16.8	1480	83.2	4
	Hormone therapy (yes)	455	12.2	113	24.8	342	75.2	4
	Targeted therapy (yes)	289	7.7	53	18.3	236	81.7	4
	Stem cell transplantation (yes)	135	3.6	15	11.1	120	88.9	4
Tumor stage	I	1595	42.7	190	11.9	1405	88.1	
	II	991	26.5	174	17.6	817	82.4	
	III	535	14.3	87	16.3	448	83.7	
	IV	171	4.6	28	16.4	143	83.6	
	Unknown	443	11.9	62	14.0	381	86.0	
Age at diagnosis—Mean (SD)		31.6 (5.9)		31.9 (5.8)		31.5 (5.9)		
Time since diagnosis—Mean (SD)		12.4 (4.5)		11.8 (4.5)		12.6 (4.5)		
	<10 years	1274	34.1	223	17.5	1051	82.5	
	10–15 years	1307	35.0	171	13.1	1136	86.9	
	>15 years	1154	30.9	147	12.7	1007	87.3	
Physical activity	Time in hours/week—Mean (SD)	27.9 (25.2)		30.1 (26.3)		27.5 (24.8)		20
	In MET hours/week—Mean (SD)	108.6 (92.3)		114.1 (96.5)		107.6 (91.4)		
BMI—Mean (SD)		25.2 (4.4)		26.9 (5.9)		24.9 (4.0)		24
	Underweight	58	1.6	11	19.0	47	81.0	
	Normal weight	2004	53.9	224	11.2	1777	88.8	
	Overweight	1218	32.8	167	13.7	1050	86.3	
	Obesity	435	11.7	136	31.3	299	68.7	
Psychosocial characteristics								
Coping style	Adaptive—Mean (SD)	29.3 (7.4)		30.0 (7.1)		29.1 (7.5)		45
	Maladaptive—Mean (SD)	13.7 (3.8)		15.7 (4.6)		12.8 (3.5)		31
HRQoL	Physical functioning—Mean (SD)	91.5 (14.0)		80.5 (18.8)		93.3 (12.1)		2
	Role functioning—Mean (SD)	83.2 (25.4)		59.7 (32.1)		87.2 (21.7)		6
	Emotional functioning—Mean (SD)	79.5 (20.6)		57.9 (24.1)		83.1 (17.5)		2
	Cognitive functioning—Mean (SD)	77.9 (24.4)		57.7 (28.5)		81.3 (21.9)		5
	Social functioning—Mean (SD)	87.9 (22.0)		66.3 (31.1)		91.6 (17.7)		13
	Global health status—Mean (SD)	75.2 (17.5)		59.2 (19.8)		77.9 (15.6)		14
Sexual attractiveness (yes)		2704	72.7	497	18.4	2203	81.6	18

^1^ Included AYA survivors completed at least half of the ten items on body image. ^2^ Digestive tract and other includes the esophagus, stomach, and small intestine. ^3^ Other includes respiratory, male genitalia (penis, prostate), urinary tract, tumor with other localizations, endocrine glands, eye, neuroblastoma, paraganglioma. ^4^ The treatments were received at primary diagnosis. ^5^ Radiotherapy includes radiotherapy for primary tumor and metastases at primary diagnosis. *n* = number of AYA survivors, Mean (SD) = mean and standard deviation.

**Table 3 cancers-14-05243-t003:** Prevalence of a negative body image among AYA survivors.

Prevalence of a Negative Body Image	AYA Survivors with a Negative Body Image		AYA Survivors without a Negative Body Image		Item Answered as ‘Not Applicable’		Missing ^1^
	*n*	%	*n*	%			*n*
Total prevalence							
Prevalence of a negative body image (with Q8)	541	14.5	3194	85.5			
Prevalence of a negative body image (without Q8)	548	14.7	3187	85.3			
Ten items on body image							
Q1. Have you felt unattractive?	391	10.5	3343	89.5			1
Q2. Have you felt older than your age?	628	16.8	3105	83.2			2
Q3. Have you been dissatisfied with your physical appearance?	603	16.2	3130	83.8			2
Q4. Have you felt that you could not trust your body?	554	14.8	3180	85.2			1
Q5. Have you felt embarrassed about your body?	297	8.0	3438	92.0			0
Q6. Did you avoid people because of the way you felt about your appearance?	96	2.6	3636	97.4			3
Q7. Did your body feel complete?	1650	44.2	2083	55.8			2
Q8. Have you been dissatisfied with the appearance of the scars? ^2^	348	9.3	3076	82.5	306	8.2	5
Q9. Did you judge your physical appearance more negatively since the diagnosis and treatment of cancer?	482	12.9	3247	87.1			6
Q10. Have you felt less masculine/feminine as a result of your illness or treatment since the diagnosis and treatment of cancer?	564	15.1	3166	84.9			5

^1^ The missing values were not part of the calculated percentages for the ten items on body image. ^2^ For item Q8 participants could also answer ‘not applicable’.

**Table 4 cancers-14-05243-t004:** Univariable and multivariable logistic regression.

Population Characteristics		Univariable Logistic Regression ^1^		Multivariable Logistic Regression ^2^	
				X^2^ = 1106.36 (*p*-Value < 0.001) Nagelkerke R^2^ = 0.47	
		OR [95%CI]	*p*-Value	OR [95%CI]	*p*-Value
Age at questionnaire		0.99 [0.98–1.01]	0.318		
Gender	Male	Reference		Reference	
	Female	3.54 [2.80–4.46]	<0.001	3.79 [2.49–5.77]	**<0.001**
Partner	Yes	Reference		Reference	
	No	2.05 [1.65–2.54]	<0.001	1.17 [0.88–1.57]	0.284
Level of education	Low	2.40 [0.94–6.09]	0.067	1.91 [0.43–8.49]	0.398
	Medium	1.51 [1.25–1.81]	<0.001	1.13 [0.88–1.44]	0.336
	High	Reference		Reference	
Type of cancer	Melanoma	Reference		Reference	
	Head and neck	1.55 [0.72–3.34]	0.260	1.68 [0.64–4.30]	0.283
	Colon and rectal	1.57 [0.65–3.76]	0.313	0.97 [0.31–2.90]	0.922
	Digestive tract other ^3^	3.33 [1.21–9.20]	0.020	3.92 [0.98–15.60]	0.053
	Breast	3.67 [2.21–6.08]	<0.001	2.51 [1.19–5.28]	**0.015**
	Female genitalia	3.89 [2.29–6.63]	<0.001	3.15 [1.59–6.24]	**<0.001**
	Thyroid gland	1.72 [0.92–3.22]	0.093	0.81 [0.36–1.83]	0.609
	Central nervous system	2.44 [1.26–4.73]	0.008	1.00 [0.33–3.03]	0.996
	Bone and soft tissue sarcoma	1.84 [0.94–3.59]	0.074	1.82 [0.77–4.29]	0.170
	Germ cell tumor	0.82 [0.46–1.47]	0.513	2.51 [1.11–5.65]	**0.027**
	Lymphoid hematological malignancies	2.12 [1.24–3.62]	0.006	2.10 [0.98–4.49]	0.055
	Myeloid hematological malignancies	2.35 [1.21–4.58]	0.012	1.78 [0.60–5.32]	0.303
	Other ^4^	2.10 [0.95–4.64]	0.069	1.20 [0.43–3.33]	0.732
Tumor stage	I	Reference		Reference	
	II	1.58 [1.26–1.97]	<0.001	1.49 [1.07–2.10]	**0.020**
	III	1.44 [1.09–1.89]	0.010	1.72 [1.13–2.61]	**0.012**
	IV	1.45 [0.94–2.23]	0.093	1.73 [0.92–3.25]	0.087
	Missing	1.20 [0.88–1.64]	0.239	1.35 [0.68–2.70]	0.390
Chemotherapy ^5^	No	Reference		Reference	
	Yes	1.40 [1.16–1.69]	<0.001	0.68 [0.47–0.99]	**0.041**
Radiotherapy ^5,6^	No	Reference		Reference	
	Yes	1.43 [1.19–1.72]	<0.001	1.12 [0.85–1.47]	0.430
Hormone therapy ^5^	No	Reference		Reference	
	Yes	2.21 [1.74–2.79]	<0.001	1.22 [0.81–1.83]	0.348
Targeted therapy ^5^	No	Reference		Reference	
	Yes	1.36 [1.00–1.86]	0.053	0.97 [0.63–1.50]	0.905
Surgery organ ^5^	No	Reference			
	Yes	1.09 [0.90–1.32]	0.398		
Surgery local ^5^	No	Reference			
	Yes	0.85 [0.65–1.12]	0.248		
Stem cell transplantation ^5^	No	Reference			
	Yes	1.37 [0.79–2.36]	0.260		
Time since diagnosis	<10 years	1.45 [1.16–1.82]	0.001	1.26 [0.94–1.69]	0.129
	10–15 years	1.03 [0.81–1.31]	0.799	0.92 [0.68–1.24]	0.570
	>15 years	Reference		Reference	
Physical activity	MET hours/week	1.00 [1.00–1.00]	0.131		
BMI	Underweight	1.86 [0.95–3.63]	0.071	0.75 [0.31–1.81]	0.516
	Normal weight	Reference		Reference	
	Overweight	1.26 [1.02–1.56]	0.034	1.70 [1.29–2.24]	**<0.001**
	Obesity	3.61 [2.82–4.61]	<0.001	3.69 [2.66–5.13]	**<0.001**
Maladaptive coping style		1.19 [1.16–1.22]	<0.001	1.10 [1.06–1.13]	**<0.001**
HRQoL	Physical functioning	0.95 [0.95–0.96]	<0.001	1.00 [0.99–1.00]	0.306
	Role functioning	0.97 [0.96–0.97]	<0.001	0.99 [0.99–1.00]	**0.009**
	Emotional functioning	0.95 [0.94–0.95]	<0.001	0.97 [0.97–0.98]	**<0.001**
	Cognitive functioning	0.97 [0.96–0.97]	<0.001	0.99 [0.99–1.00]	**0.006**
	Social functioning	0.96 [0.96–0.97]	<0.001	0.99 [0.98–1.00]	**0.001**
	Global Health status	0.95 [0.94–0.95]	<0.001	0.98 [0.97–0.99]	**<0.001**
Sexual attractiveness	No	Reference		Reference	
	Yes	5.66 [4.05–7.91]	<0.001	3.73 [2.50–5.57]	**<0.001**

^1^ Univariable: *p*-value < 0.1 is included in multivariable analyses. ^2^ Multivariable: *p*-value < 0.05 is significant (bold *p*-values show a statistically significant OR). Method = enter. The multivariable model showed no multicollinearity. ^3^ Digestive tract and other includes the esophagus, stomach, and small intestine. ^4^ Other includes respiratory, male genitalia (penis, prostate), urinary tract, tumor with other localizations, endocrine glands, eye, neuroblastoma, paraganglioma. ^5^ The treatments were received at primary diagnosis. ^6^ Radiotherapy includes radiotherapy for primary tumor and metastases at primary diagnosis. OR [95%CI] = odds ratio and 95% confidence interval.

## Data Availability

The data presented in this study are available on request from the corresponding author. The data are not publicly available due to privacy issues.

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
