# Peer review of "A Negative Body Image among Adolescent and Young Adult (AYA) Cancer Survivors: Results from the Population-Based SURVAYA Study"

_cancers, 2022, doi:10.3390/cancers14215243_

Round 1

Reviewer 1 Report

Dear authors,

thank you very much for this manuscript on a very important topic. I have the following questions/ comments:

1. results: Do I understand the first sentence correctly? Do the 36 % response percentage equal the 4010 survivors? Or what was the total number within the SURVAYA study?

2. table 1: what is the difference between surgery organ and surgery local? Type stem cell transplantation. Tumor stage: how did you classify the CNS tumor within that?

3. table 2: the physical activity seems to be high - definition? Any explanations?

4. Are there any differences regarding treatment period/ time since diagnosis since surgical techniques and treatment has changed a lot over the last decades?

5. What are now your concrete steps for the AYA population in the Netherlands based on your results?

Reviewer 2 Report

This is a well written article providing useful quantitative data on the topic of negative body image in AYA cancer survivors. As such, it is a useful addition to the current body of knowledge.

Sample size was good although on the downside the response rate was low (36%) making results less reliable and open to selection bias. Also participants were not required to answer all the questions.  There was a gender imbalance in respondents (61% female) which may have skewed the results. 

There is no way to compare results of this survey to body image problems in the general population so it is difficult to acsertain whether this is any better or worse than the general population.  

Reviewer 3 Report

The Authors present a paper:" A negative body image among Adolescent and Young adult (AYA) Cancer Survivors: Results from the population-based SURVAYA study" well written and documented but not original. The negative implications for the self-esteem, self-identity, and social relationship of AYAs due to body changes, skin burns and physical movement limitations are already well reported in the literature even if  in their study are related to different factors: age, gender, time since diagnosis, physical activity level, Body mass index, coping style, HRQoL and sexual attractiveness.

The reported negative body image in survivors is low (15%) compared to the total number of cured subjects. The low percentage is acceptable in front of recovering the life (cure).

No mention about contemporary reconstructive surgery in breast cancer surgery.

No mention about the percentage of higher BMI or maladaptive coping styles with the risk for having a negative body image in thermal population (not affected by cancer). This comparison should be interesting.

What is  CONCRETELY the  important contribution and insight into the long-term impact of cancer and several related factors with a negative body image?

Please indicate the type of psychosocial support for AYA survivors who experience a negative body image do you suggest or provide. 

Round 2

Reviewer 3 Report

I liked the effort of Authors to answer the requests of reviewers clarifying in a comprehensive way the points arose by reviewer.

At the present in this specific shape I don't have further request to them

Author Response

Response to Reviewer 3 comments:

I liked the effort of Authors to answer the requests of reviewers clarifying in a comprehensive way the points arose by reviewer.

At the present in this specific shape I don't have further request to them

Response: We thank the reviewer for the reply to the revisions.